# Iron Deficiency in Vegetarian and Omnivorous Individuals: Analysis of 1340 Individuals

**DOI:** 10.3390/nu13092964

**Published:** 2021-08-26

**Authors:** Eric Slywitch, Carine Savalli, Antonio Cláudio Goulart Duarte, Maria Arlete Meil Schimith Escrivão

**Affiliations:** 1Paulista School of Medicine, Federal University of São Paulo (Unifesp), Sao Paulo 04021-001, Brazil; maria.arlete@uol.com.br; 2Department of Public Policy and Public Health, Federal University of São Paulo (Unifesp), Sao Paulo 04021-001, Brazil; carine.savalli@unifesp.br; 3Medical School, Federal University of Rio de Janeiro (UFRJ), Rio de Janeiro 21941-901, Brazil; antonioclaudio@ufrj.br

**Keywords:** iron, iron deficiency, obesity, insulin resistance, inflammation, vegetarian diet, plant-based diet, omnivorous diet

## Abstract

The objective of this study was to evaluate the serum levels of ferritin and the prevalence of iron deficiency in vegan and omnivorous individuals by taking into account the presence of elements that cause an elevation of ferritin levels, such as increased homeostatic model assessment of insulin resistance (HOMA-IR), body mass index (BMI), and high-sensitivity C-reactive protein (hs-CRP) values. The parameters were evaluated in 1340 individuals, i.e., 422 men and 225 women who do not menstruate and 693 women who do menstruate, based on omnivorous or vegetarian eating habits. The progressive increase in BMI, HOMA-IR, and inflammation caused an elevation in ferritin concentration, regardless of the eating habits in the groups studied. In the overall sample, omnivores had a higher prevalence of obesity, higher ferritin levels, and a lower prevalence of iron deficiency (ferritin < 30 ng/mL). However, after the exclusion of individuals with inflammation (with overweight/obesity and elevated hs-CRP levels), the actual iron deficiency was assessed and was not higher among vegetarians, except in women with regular menstrual cycles. Our data show that nutritional status and inflammation levels affect ferritin levels and may interfere with the correct diagnosis of iron deficiency in both vegetarian and omnivorous individuals. Compared to vegetarians, women who do not menstruate and men had the same prevalence of iron deficiency when following an omnivorous diet.

## 1. Introduction

Iron deficiency negatively impacts organ functions and is associated with an increased risk of maternal and child mortality, impaired cognitive and body development, reduced physical performance and work capacity in adults, and decreased cognitive function in elderly individuals [1]. Dietary composition can affect body iron stocks because the intake of iron in the heme form has higher bioavailability than in the nonheme form. The absorption of the latter is favored (vitamin C, citric acid, and other organic acids) or inhibited (phytates, polyphenols and calcium) by dietary elements [2]. Approximately 10–15% of heme iron is available in an omnivorous diet rich in meat; notably, this iron is excluded from the vegetarian diet [3].

Observational studies suggest a protective effect of a vegetarian diet on the development of several chronic noncommunicable diseases, including a reduced risk of ischemic heart disease (25%) and cancer (8–15%) [4]. Compared with omnivorous habits, vegetarian habits are also more effective in the glycemic control of diabetic individuals [5,6,7]. However, some studies report a higher prevalence of iron deficiency (with or without anemia) in vegetarian populations, especially in women [8,9,10].

The first variable affected by iron deficiency is ferritin, an acute-phase protein that mostly originates from hepatocytes but that can also be produced by other cells, such as macrophages [11].

Serum ferritin concentrations correlate with body iron stocks, where values below 30 ng/mL are considered the most sensitive (92%) and specific (98%) markers of iron deficiency [12]. However, even in the presence of low iron stores, some conditions can increase the ferritin level and constitute confounding factors in the diagnosis of iron deficiency, such as inflammation and infection [1]. In this scenario, the evaluation of iron deficiency depends on the simultaneous measurement of ferritin and other important inflammatory markers, such as high-sensitivity C-reactive protein (hs-CRP), alpha-1-acid glycoprotein, and TNF-alpha [13,14].

Several factors can trigger inflammation in the absence of an acute inflammatory or infectious trigger, including excess body fat [15]. Body fat accumulation is considered an underlying mechanism of metabolic diseases because it triggers low-grade inflammation and insulin resistance [16], and both conditions increase ferritin synthesis [17]. Lower body mass index (BMI) values are reported in individuals with vegetarian habits than in omnivores [18]. However, comparative studies between these populations do not include correlations of ferritin concentrations with BMI or other inflammatory factors that can affect the metabolism when determining whether the lower ferritin levels reported in vegetarians may be due to a more compromised iron nutritional status or to lower metabolic inflammation.

The objective of our study was to investigate the effect of the determinants of ferritin metabolism on circulating ferritin concentrations and to evaluate iron deficiency in vegetarian and omnivorous individuals. The parameters were evaluated in men, women who menstruate, and women who do not menstruate.

## 2. Materials and Methods

### 2.1. Ethical Issues

The present retrospective, cross-sectional, and observational study was approved by the Research Ethics Committee of the Federal University of São Paulo/Paulista School of Medicine (UNIFESP/EPM) under CEP n. 1.052.082. All participants signed an informed consent form.

### 2.2. Sample

Data from patients aged 18 to 60 years of both sexes treated at a private clinic in the city of São Paulo (Brazil) between 2008 and 2018 were included in this study. Pregnant or nursing women; patients using medications or supplements in the last 6 months able to influence the evaluated variables; patients undergoing nutritional intervention in the last 6 months; smokers; patients with alcohol consumption > 3 times per week; practitioners of physical activity > 3 times a week; patients diagnosed with diseases capable of influencing the evaluated variables (thalassemia, liver disease, hypo- or hyperthyroidism, inflammatory bowel disease, hemochromatosis, cancer with possible metabolic repercussions, rheumatic and autoimmune diseases with systemic inflammatory responses, irritable bowel syndrome and gastrointestinal tract surgeries that affect nutrient absorption (total or partial gastrectomy, gastric bypass, gastroduodenopancreatectomy and bariatric surgery); patients of Asian ethnicity; and patients who donated blood in the last 12 months were not included in the study. Ultimately, 1340 individuals were included, namely 422 men and 918 women; among the women, 226 did not menstruate, and 691 had regular menstrual cycles (for one of the women, there was no menstrual cycle information).

### 2.3. Sample Classification According to Eating Habits

The selected individuals were classified into four groups based on food choices in the last 12 months: (1) omnivores (individuals who theoretically eat all food groups), (2) semi-vegetarians (individuals who eat white meat in up to 3 meals per week), (3) lacto-ovo vegetarians (individuals who do not eat any type of meat but eat eggs, milk, and dairy products), and (4) vegans (individuals who, in addition to abstaining from meats, do not eat any animal products, such as eggs and dairy products). The grouping of semi-vegetarians, lacto-ovo vegetarians, and vegans into a single group called “vegetarian” was subsequently considered for comparative purposes with the omnivore group.

### 2.4. Sample Classification Based on Nutritional Status

The nutritional status of the individuals studied was determined from their BMI, in accordance with the criteria proposed by the World Health Organization [19]: low weight = BMI < 18.5; normal weight = BMI ≥ 18.5 and <25.0; overweight = BMI ≥ 25.0 and <30.0; obesity (grade I) = BMI ≥ 30.0 and <35.0; obesity (grade II) = BMI ≥ 35 and <40; and obesity (grade III) = BMI ≥ 40. The BMI was obtained by dividing the weight (in kg) by the squared height (in meters). Body weight was measured with the patient barefoot and wearing only undergarments and standing in the center of a Filizola electronic scale with a 150-kg capacity and 100-g accuracy; height was measured using an Alturexata stadiometer (0.1-cm accuracy) with the patient standing barefoot with the heels together, the back erect, and the arms extended at the side of the body.

### 2.5. Analysis of Markers of Iron Metabolism and Metabolic Inflammation

We evaluated the serum concentrations of hs-CRP and ferritin and the glucose profile (fasting glucose and insulin). These biochemical values were obtained from blood samples that were collected and analyzed in laboratories with an international quality certification issued by at least one of the following accreditors: ISO 9001, ISO 14001, PALC, the College of American Pathologists (CAP, Northfield Township, IL, USA), CAP Accredited^TM^, PALC/SBPC/ML, the American Association of Blood Banks (AABB), the Foundation for the Accreditation of Cellular Therapy (FACT, Seattle, WA, USA), the Joint Commission International, OHSAS 18001, the Commission on the Accreditation of Rehabilitation Facilities (CARF), HIMSS Analytics, and the Joint Commission International (JCI, Oakbrook Terrace, IL, USA). The laboratory used for analysis was chosen by the patient. Hs-CRP values were applied to evaluate the degree of inflammation, categorized as low (≤0.1 mg/dL), moderate (0.1 to 0.3 mg/dL), or high (>0.3 mg/dL), and glucose profile values were used to calculate the homeostatic model assessment-insulin resistance (HOMA-IR) index, where HOMA-IR = (fasting insulin [mcIU/mL] × fasting glucose [mmol/L])/22.5.

### 2.6. Statistical Analysis

The chi-squared test was used to compare the groups (men, women who do not menstruate, and women who menstruate) regarding the distribution of the four types of eating habits, and a partitioned chi-squared analysis was used to identify the differences in detail. To compare the nutritional status among the various diets in each group (men and women), the chi-squared test and partitioned chi-squared analysis were also used. One-way analysis of variance was used to compare the groups (men, women who do not menstruate, and women who menstruate) relative to BMI, hs-CRP and HOMA-IR.

Two-factor analysis of variance was used to compare the groups (men, women who do not menstruate, and women who menstruate) and eating habits relative to ferritin concentrations, including the study of interactions. To include the degree of inflammation in this analysis, an analysis of variance model with a third factor, hs-CRP in three categories (low, moderate and high inflammation), was used. For this analysis, individuals with vegetarian eating habits (lacto-ovo vegetarian, semi-vegetarian, and vegan) were grouped and compared with omnivores. The semi-vegetarians were grouped with the vegetarians and not with the omnivores because meat consumption was low in this group (up to 3 meals per week), because white meat has a lower iron content, and because previous studies have performed this grouping.

The relationships between ferritin and BMI and ferritin and HOMA-IR, based on the groups (men, women who do not menstruate and women who menstruate) and eating habits, were investigated using linear regression models with the logarithm of the variables. The effect of the group (men, women who menstruate, or not) and degree of inflammation (hs-CRP in 3 categories: low, moderate, and high) on the HOMA-IR levels was analyzed using two-factor analysis of variance. The chi-squared test (or Fisher’s exact test when more appropriate) was used to compare the proportions of individuals with altered ferritin (<30 µg/L) between the omnivorous and vegetarian eating habits in each group. This analysis also grouped the vegetarian eating habits due to the limited sample size. In a second step, the same analyses were repeated considering only one subgroup of the sample (BMI between 18.5 and 24.9 kg/m^2^ and hs-CRP values < 0.10).

Because the methods used assume normality for the quantitative data, it was necessary to apply logarithmic transformation to ensure this assumption. For all models, the goodness-of-fit and assumption testing were performed through residual analysis. All of the results were interpreted using a significance level of 5%, and the Bonferroni correction for multiple comparisons was used. The calculations were performed using the SAS (Statistical Analysis System) software package.

## 3. Results

### 3.1. Descriptive Characteristics of the Overall Sample (without Excluding Individuals with Inflammation or Insulin Resistance, Factors That Alter Ferritin Levels)

The mean age ± standard deviation of the subjects included in this study was 37.5 ± 9.8 years. Of the total, 422 were men (37.4 ± 9.3 years, 18–60 years), 226 were women who had not menstruated for at least 1 year, either due to the use of contraceptive methods or to being postmenopausal (47.3 ± 10.1 years, 22–60), and 691 were women who had regular menstrual cycles (34.3 ± 7.6, 18–53). The comparisons among the groups of men, women who do not menstruate, and women who menstruate showed heterogeneity in the sample distribution in relation to eating habits, with a higher prevalence of lacto-ovo vegetarians in the group of women who menstruate and a higher prevalence of omnivores in the group of women who do not menstruate (Figure 1).

To comparatively evaluate the groups (men, women who do not menstruate, and women who menstruate) regarding BMI, hs-CRP, and HOMA-RI, analysis of variance was used (Table 1). Women who menstruate had a lower BMI than those who do not menstruate (F_(1,1336)_ = 28.50, *p* < 0.0001) and a lower HOMA-IR than men and women who do not menstruate (F_(1,1241)_ = 29.00, *p* < 0.0001 and F_(1,1336)_ = 12.31, *p* = 0.0005, respectively). Men had significantly lower hs-CRP than the other groups (F_(1,1235)_ = 12.13, *p* = 0.0005 when compared to women who menstruate, and F_(1,1235)_ = 9.13, *p* = 0.0026 when compared with women who do not menstruate). Ferritin was significantly different among the three groups, and men had higher ferritin levels than women who do not menstruate (F_(1,1307)_ = 1054.52, *p* < 0.0001); the latter, in turn, had higher ferritin levels than women who menstruate (F_(1,1307)_ = 159.84, *p* < 0.0001).

### 3.2. Vegetarian Habits Were Associated with a Lower Prevalence of Obesity and Lower Serum Ferritin Levels, Regardless of Inflammation

The groups of women who do not menstruate and who do menstruate were grouped for this analysis. Eating habits were associated with nutritional status for both men (X^2^ = 37.5515; df = 9; *p* ≤ 0.0001) and women (X^2^ = 47.5992; df = 9; *p* ≤ 0.0001) (Table 2). The partitioned chi-squared analysis showed that in men, the prevalence of obesity was lower in lacto-ovo vegetarians (*p* = 0.0004) and semi-vegetarians (*p* = 0.0028) but not in vegans (*p* = 0.0794) when compared to omnivores, whereas in women, the prevalence of obesity was lower in all women with vegetarian habits than in omnivores (omnivore vs. lacto-ovo vegetarian, *p* < 0.0001; omnivore vs. semi-vegetarian, *p* = 0.0137; omnivore vs. vegan, *p* = 0.0045). In addition, the frequency of low weight was higher in vegan women than in women who had other types of eating habits. The data on nutritional status based on eating habits are provided in Table 2.

Sex (F_(2,2298)_ = 383.5900, *p* < 0.0001) and eating habit (F_(3,1298)_ = 45.7600, *p* < 0.0001) affected serum ferritin concentrations, with a borderline interaction that did not reach the 5% significance level (F_(6,1298)_ = 2.0300, *p* = 0.0585). However, we continued the analyses considering the possibility that the effect of eating habits may be different for the sexes. Higher mean serum ferritin levels were found in men, followed by women who do not menstruate and women who menstruate for all eating habits (*p* ≤ 0.0062). However, among men and women who menstruate, lower serum ferritin was found in vegetarians than in omnivores (*p* ≤ 0.0002) and for women who menstruate; the means for this variable were similar among the groups of vegetarian women (*p* ≥ 0.6833). Among women who do not menstruate, those who were lacto-ovo vegetarian (*p* < 0.0001) and semi-vegetarian (*p* = 0.0225) had lower mean ferritin levels than omnivores did. Although the last result did not reach the corrected significance level for multiple comparisons and should be considered only as a trend, similar mean concentrations of this protein were observed between vegetarian and omnivorous women (*p* = 0.1754) and the entire vegetarian population (*p* ≥ 0.0615).

To include the effect of hs-CRP in this analysis, it was necessary to group the vegetarian diets because of the limited sample size. When distributing the sample based on the degree of inflammation (using hs-CRP values as a reference), the vegetarian individuals had lower ferritin concentrations than the omnivores did, regardless of sex and menstrual blood loss (Figure 2).

Among the different degrees of inflammation, vegetarians had lower ferritin levels than omnivores did, regardless of sex and menstrual blood loss (*p* ≤ 0.0023), except for women who do not menstruate, who presented with a low degree of inflammation (*p* = 0.2633) and for women who menstruate, who presented with a moderate degree of inflammation (*p* = 0.1343). 

### 3.3. In the Overall Sample (without Excluding Individuals with Inflammation or Insulin Resistance), the Frequency of Iron Deficiency Was Higher among Vegetarians Than Omnivores, Regardless of Sex and Menstrual Blood Loss

Regarding the diagnosis of iron deficiency in the overall sample (*n* = 1310), the frequency of iron deficiency (ferritin < 30 μg/L) was higher in women who menstruate (41.78%) than in those who do not menstruate (10.31%) and in men (1.70%) and was 4.0 times higher in women who menstruate than in women who do not (X^2^_(df = 2)_ = 253.5001, *p* < 0.0001). Eating habits were associated with iron deficiency, which was more prevalent in vegetarian than in omnivorous men and in vegetarian than in omnivorous women, regardless of whether they menstruated (Table 3).

### 3.4. Serum Ferritin Concentrations Had a Positive Relationship with Increasing BMI and HOMA-IR Values Associated with Inflammation, Regardless of Eating Habit and Sex

In a second-order interaction model, eating habit (*p* = 0.1266), sex (*p* = 0.4456), and the interaction between these variables (*p* = 0.6688) did not affect the relationship between BMI and serum ferritin. An estimated 1.7% increase in serum ferritin was observed for each 1.0% increase in BMI, regardless of eating habit and sex (Figure 3A). Similarly, in a second-order interaction model, the relationship between HOMA-IR values and serum ferritin concentrations was not affected by eating habits (*p* = 0.1193), sex (*p* = 0.3567) or the interaction between these variables (*p* = 0.0987). An estimated increase of 0.4% in serum ferritin was observed for each 1.0% increase in HOMA-IR, regardless of eating habits or sex (Figure 3B). Figure 3 shows that increases in serum ferritin concentrations occurred in parallel to the increase in (A) BMI (*p* = 0.0086) and (B) HOMA-IR (*p* = 0.0010) values.

The increase in HOMA-IR values was associated with an increase in the degree of inflammation, as determined by hs-CRP levels, regardless of sex (Figure 4). Figure 4 indicates that HOMA-IR values increase in parallel to the degree of inflammation, regardless of sex or menstrual blood loss.

### 3.5. Considering Only Individuals without Inflammation or Insulin Resistance, Enabling a True Diagnosis, Iron Deficiency Was the Same in Men and Women Who Do Not Menstruate, Regardless of Eating Habit, and More Prevalent Only in Vegetarian Women Who Menstruate

An analysis including only patients with a BMI between 18.5 and 24.9 kg/m^2^ and hs-CRP values < 0.10 (*n* = 773) revealed a higher frequency of iron deficiency in women who menstruate (44.85%) than in women who do not menstruate (9.57%) and men (2.08%) and a higher frequency in women who menstruate than in women who do not menstruate (X^2^_(df = 2)_ = 145.5769; *p* < 0.0001). Regarding the effect eating habits on iron deficiency, the aforementioned analysis did not find differences in prevalence between vegetarian and omnivorous men (*p* = 0.3032) or between vegetarian and omnivorous women who do not menstruate (*p* = 0.1062); however, among women who menstruate, compared to omnivores, vegetarians had a higher prevalence of iron deficiency (*p* < 0.0001) (Table 4).

## 4. Discussion

Our study confirmed previous observations that reported higher circulating ferritin concentrations in omnivorous than vegetarian individuals among both men and women [20]. Although metabolic inflammation is associated with elevated levels of this acute-phase protein not reflecting the nutritional status of iron in this condition, the influence of factors able to modify ferritin metabolism in vegetarians and omnivores was not evaluated in those studies. The exception is the study by Haddad et al., who evaluated CRP concentrations and found similar values of this inflammatory marker among vegetarian and omnivorous individuals, and despite finding significantly higher BMI values in omnivores, these data were not considered in the comparative evaluation of ferritin levels [21].

Thus, we do not know whether the more elevated ferritin levels in omnivores mentioned in those studies indicate the nutritional status of iron or a higher inflammatory state of the individual.

Recently, through the measurement of hs-CRP, two meta-analyses showed that compared to omnivores, vegetarians have significantly lower levels of inflammation [22,23].

Compared to omnivores, vegetarians have a more preserved antioxidant state and, consequently, lower metabolic inflammation and greater insulin sensitivity [7], which could explain the lower ferritin levels.

Elevated ferritin is independently associated with metabolic syndrome and its hepatic manifestation, nonalcoholic fatty liver disease [24,25]. Excess body weight, which is common in this syndrome, may contribute to peripheral insulin resistance (PIR) and hepatic fat accumulation by the mechanisms involving inflammation [24,26,27]. A previous study showed that ferritin concentrations in individuals with overweight and obesity are a marker of inflammation and not of iron nutritional status [28]. In our study, compared to omnivorous habits, vegetarian habits were associated with a lower prevalence of obesity (determined by BMI values) and lower circulating concentrations of ferritin, a finding similar to that reported by previous studies [21,29]. We found that increased inflammation levels and weight gain elevate ferritin similarly in both omnivores and vegetarians. This observation suggests that higher levels of ferritin observed in omnivorous individuals may have been influenced by the higher prevalence of overweight/obesity in this population than in vegetarian individuals.

Likewise, a study comparing vegetarian and omnivorous individuals with low weight/normal weight (BMI < 23) also found lower mean ferritin levels in vegetarians (lacto-ovo vegetarians) than in omnivores (35 μg/L vs. 72 μg/L, respectively) [30]. Interestingly, omnivores had lower insulin sensitivity than vegetarians did, but this increased 40% after therapeutic bloodletting, which aimed to achieve ferritin levels similar to those of vegetarians. In our study, increases in HOMA-IR values were related to increases in the degree of inflammation, determined by ranges of circulating hs-CRP levels. In this sense, the composition of the omnivorous diet may have a proinflammatory effect. For example, meat (especially red) and eggs supply choline for the synthesis of trimethylamine (TMA) by the intestinal microbiota, a molecule absorbed by the intestine and that is oxidized by the liver to trimethylamine N-oxide (TMAO), which is associated with a greater risk of inflammatory metabolic changes, including insulin resistance [31]. Together, these observations suggest that the relationship between increased HOMA-IR and elevated ferritin levels may be independent of overweight/obesity and that omnivorous habits may promote such increases, possibly due to the inflammatory mechanisms also resulting from the effect of the consumption of a greater amount of saturated fat [32] and because foods of animal origin contain very low amounts of antioxidants [33].

Therefore, the lower ferritin concentrations in vegetarian than in omnivorous individuals may not reflect significant differences in the prevalence of iron deficiency between these populations. In some cases, very high levels of this protein (above 200 ng/mL) can be found in omnivores [34] with harmful potential because it is associated, for example, with a higher risk of hyperuricemia [35], cell damage by excess free iron [36], cytotoxic effects on colonocytes (by the presence of heme iron), increased lipid peroxidation, increased endogenous nitrosation, and negative changes in the human microbiome, increasing the risk of colorectal cancer [37].

Importantly, different concentrations of ferritin were used in the studies, as they reflect a higher or lower prevalence of iron deficiency among them. There have been two studies that have considered a ferritin cutoff point of <12 ng/mL and that found a similar prevalence of iron deficiency between vegetarian and omnivorous women, including between those who menstruated and those who did not menstruate in one of those studies [38,39]. A study that considered different ferritin concentration cutoff points for men (<20 ng/mL) and women (<15 ng/mL) found a similar prevalence of iron deficiency between vegetarian and omnivorous men but a higher prevalence of iron deficiency in vegetarian (lacto-ovo vegetarian) women than in omnivorous women [40]. Another study that considered a ferritin cutoff point of <25 ng/mL found a higher frequency of iron deficiency in men with different vegetarian habits (lacto-ovo vegetarians and vegans) than in omnivores [34]. None of these studies evaluated the effect of metabolic and/or inflammatory factors determinant of ferritin metabolism on the diagnosis of iron deficiency.

In our study, we used ferritin < 30 ng/mL for the diagnosis of iron deficiency. Without filtering the factors that affect ferritin levels, such as BMI and hs-CRP values, vegetarian individuals had a higher frequency of iron deficiency than omnivores did, regardless of sex and menstrual blood loss. However, when individuals with overweight/obesity and inflammation were excluded from the analysis, to use ferritin as a marker of iron nutritional status, this observation only remained true for menstrual blood loss.

The present study showed that factors involved in ferritin metabolism may influence its serum concentration and may constitute confounding biases, especially BMI and PIR associated with inflammation, in the assessment of the prevalence of iron deficiency. When we excluded these factors from the analysis, we found that there was no difference in the prevalence of iron deficiency in the vegetarian and omnivorous participants who do not lose blood (men and women who do not menstruate). Vegetarian habits, compared to omnivores habits, may contribute to greater iron deficiency only in women with menstrual blood loss. In this sense, blood loss is the most important factor for the occurrence of this condition. Therefore, it seems important that vegetarian individuals, in comparison to omnivorous individuals, should pay increased attention to their iron nutritional status when they show blood loss of any origin by choosing foods with higher concentrations of iron, optimizing factors that improve its absorption, and avoiding those that hinder it.

The main limitation of this study is the lack of an evaluation of the consumption of dietary iron. However, the representativeness of this variable may be weak because the type of iron consumed and different dietary components influence the bioavailability and absorption of this mineral and are difficult to control [2]. We also found a difference between the proportional distribution of women who menstruate and women who do not menstruate within the different eating habits. However, intergroup comparisons by eating habits encompassed a significant number of individuals in each group, enabling reliable results despite the differences in the proportional distribution of individuals between them.

The sample size was one of the strengths of our study. While a meta-analysis involving 24 cross-sectional studies that comparatively evaluated serum ferritin levels between vegetarians and omnivores and that included both sexes totaled slightly more than 2100 individuals [20], our sample consisted of 1340 individuals. In addition, we evaluated the influence of factors that may affect circulating ferritin concentrations and, consequently, the diagnosis of iron deficiency, which has not yet been assessed between vegetarian and omnivorous populations. We also demonstrated that omnivores and vegetarians present a similar increase in ferritin levels with weight gain and increased PIR. These evaluations were performed in a sample not only of significant size but that also included both sexes and that allowed comparisons between individuals with and without menstrual blood loss.

For future studies, we point to the need to evaluate the levels of alpha-1-acid glycoprotein, TNF-alpha, uric acid, homocysteine, and the erythrocyte sedimentation rate (ESR) between vegetarian and nonvegetarian groups, as these are elements that can also alter ferritin levels. We do not yet know whether the elevated ferritin levels in the omnivore group were due to higher amounts of body iron or a higher inflammatory status due to more elevated biomarkers in omnivores because a dietary profile with higher saturated fat and less fiber content may favor these changes [32].

## 5. Conclusions

Our data indicate that the nutritional status of iron may be masked by body inflammation. The evaluation of ferritin should always be performed in conjunction with inflammatory markers (hs-CRP), BMI, and insulin resistance, both in vegetarian and omnivorous individuals.

We did not find differences in the prevalence of iron deficiency between vegetarians and omnivores among men and women who do not menstruate. Among women who menstruate, vegetarians had a higher prevalence of iron deficiency than omnivores did. It is possible that this risk extends to individuals with significant blood loss of any nature.

## Figures and Tables

**Figure 1 nutrients-13-02964-f001:**
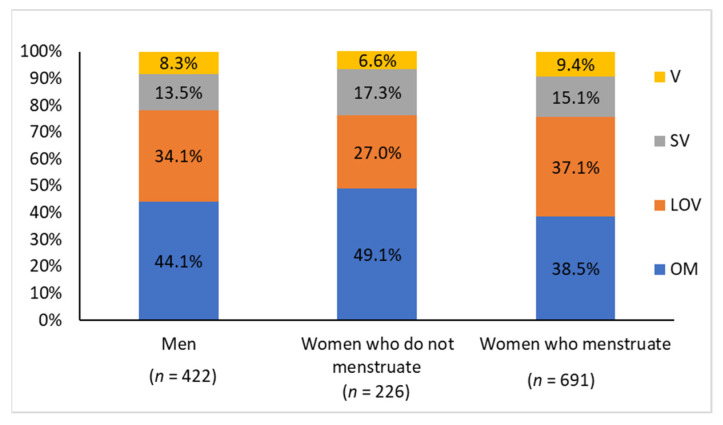
Sample distribution in relation to eating habits for each group: OM (omnivore), LOV (lacto-ovo vegetarian), SV (semi-vegetarian), and V (vegan). Comparison of the distribution of eating habits between groups: men vs. women who do not menstruate (*p* = 0.1645); men vs. women who menstruate (*p* = 0.3306); women who do not menstruate vs. women who menstruate (*p* = 0.0086).

**Figure 2 nutrients-13-02964-f002:**
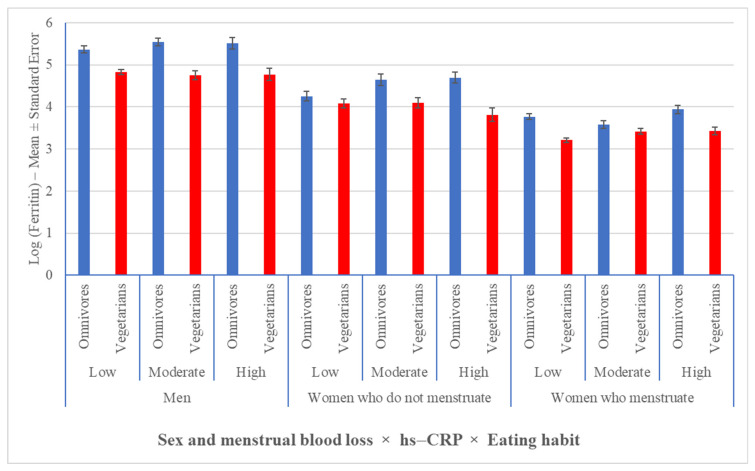
Variation in serum ferritin levels in vegetarian and omnivorous individuals based on degrees of inflammation, sex, and menstrual blood loss. Data expressed on the logarithmic scale.

**Figure 3 nutrients-13-02964-f003:**
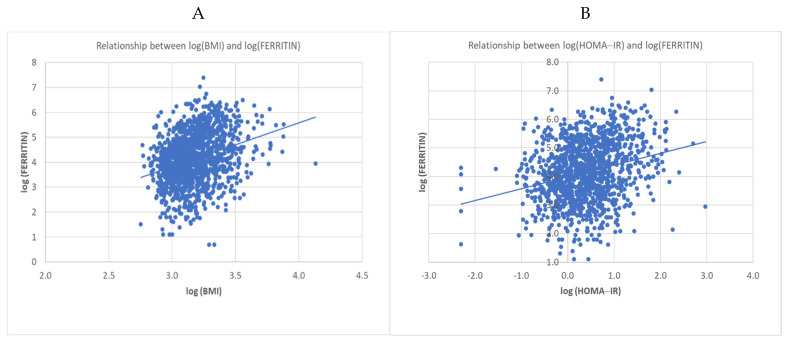
(**A**) Scatter plot of ferritin vs. body mass index (BMI), both on the logarithmic scale (Pearson’s correlation = 0.30; *p* < 0.0001; *n* = 1311). (**B**) Scatter plot of ferritin vs. homeostatic model assessment-insulin resistance (HOMA-IR), both on the logarithmic scale (Pearson’s correlation = 0.26; *p* < 0.0001; *n* = 1241).

**Figure 4 nutrients-13-02964-f004:**
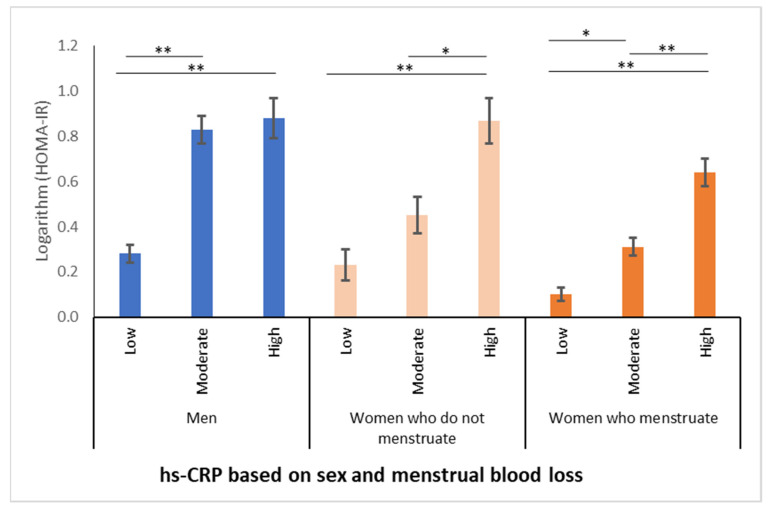
Relationship between homeostatic model assessment-insulin resistance (HOMA-IR) values and degrees of inflammation in men and women (all eating habits grouped). * *p* ≤ 0.001; ** *p* < 0.0001. Data expressed on the logarithmic scale.

**Table 1 nutrients-13-02964-t001:** Descriptive parameters of the study groups.

Variable (Mean ± SD)	Men A	Women Who Do Not Menstruate B	Women Who Menstruate C	*p*-Value (A vs. B)	*p*-Value (A vs. C)	*p*-Value (B vs. C)
BMI (kg/m^2^)	25.51 ± 4.4	25.17 ± 5.6	23.32 ± 4.3	0.1551	<0.0001	<0.0001
hs-CRP (mg/dL)	0.61 ± 2.1	0.85 ± 3.0	0.96 ± 3.7	0.0026	0.0005	0.6829
HOMA-IR	2.14 ± 1.6	2.10 ± 1.9	1.61 ± 1.2	0.4477	<0.0001	0.0005
Ferritin	208.88 ± 162.29	89.35 ± 72.03	45.79 ± 39.59	<0.0001	<0.0001	<0.0001

Although shown in the original scale, the BMI, hs-CRP, HOMA-IR, and ferritin variables were compared on the logarithmic scale due to strong positive asymmetry. To interpret the *p*-values, consider the multiple comparisons correction (significance level of 1.67%). SD, standard deviation; HOMA-IR, homeostatic model assessment of insulin resistance; BMI, body mass index; CRP, C-reactive protein.

**Table 2 nutrients-13-02964-t002:** Distribution of nutritional status based on eating habits.

Eating Habit	Nutritional Status—Men					Nutritional Status—Women				
	Low Weight *n* (%)	Eutrophy *n* (%)	Overweight *n* (%)	Obesity *n* (%)	Total	Low Weight *n* (%)	Eutrophy *n* (%)	Overweight *n* (%)	Obesity *n* (%)	Total
Omnivore	2 1.1%	67 36.0%	75 40.3%	42 22.6%	186 100.0%	13 3.5%	225 59.5%	75 19.8%	65 17.2%	378 100.0%
Semi-vegetarian	1 1.8%	35 61.4%	20 35.1%	1 1.7%	57 100.0%	9 6.3%	96 67.1%	31 21.7%	7 4.9%	143 100.0%
Lacto-ovo vegetarian	8 5.6%	76 52.8%	48 33.3%	12 8.3%	144 100.0%	17 5.4%	224 70.6%	60 18.9%	16 5.1%	317 100.0%
Vegan	2 5.7%	20 57.1%	10 28.6%	3 8.6%	35 100.0%	11 13.8%	49 61.2%	13 16.3%	7 8.7%	80 100.0%
Total	13	198	153	58	422	50	594	179	95	918

**Table 3 nutrients-13-02964-t003:** Overall sample: diagnosis of iron deficiency among men and women who do not menstruate and menstruate based on eating habits.

Sample	Habit	Ferritin		Total	*p*-Value
		Normal (≥30 µg/L)	Altered (<30 µg/L)		
Men	Vegetarian	225	7	232	0.0201 ^(2)^
	%	96.88	3.02	100.00	
	Omnivore	180	0	180	
	%	100.00	0	100.00	
	Total	405	7	412	
Women who do not menstruate	Vegetarian	95	19	114	0.0014 ^(1)^ X^2^_(df = 1)_ = 10.1756
	%	83.33	16.67	100.00	
	Omnivore	105	4	109	
	%	96.33	3.67	100.00	
	Total	200	23	223	
Women who menstruate	Vegetarian	214	206	420	<0.0001 ^(1)^ X^2^_(df = 1)_ = 24.1562
	%	50.95	49.05	100.00	
	Omnivore	179	76	225	
	%	70.20	29.80	100.00	
	Total	257	209	675	

^(1)^ Chi-squared test; ^(2)^ Fisher’s exact test.

**Table 4 nutrients-13-02964-t004:** Diagnosis of iron deficiency among men and women who do not menstruate and menstruate based on eating habits.

Sample	Habit	Ferritin		Total	*p*-Value
		Normal (≥30 µg/L)	Altered (<30 µg/L)		
Men	Vegetarian	124	4	128	0.3032 ^(2)^
	%	96.88	3.13	100.00	
	Omnivore	64	0	64	
	%	100.00	0	100.00	
	Total	188	4	192	
Women who do not menstruate	Vegetarian	49	8	57	0.1062 ^(1)^ X^2^_(df = 1)_ = 2.6104
	%	85.96	14.04	100.00	
	Omnivore	55	3	58	
	%	94.83	5.17	100.00	
	Total	104	11	115	
Women who menstruate	Vegetarian	150	159	309	<0.0001 ^(1)^ X^2^_(df = 1)_ = 16.1839
	%	48.54	51.46	100.00	
	Omnivore	107	50	157	
	%	68.15	31.85	100.00	
	Total	257	209	466	

^(1)^ Chi-squared test, ^(2)^ Fisher’s exact test.

## Data Availability

The data presented in this study are available on request from the corresponding author.

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
