# Peer review of "Iron Deficiency in Vegetarian and Omnivorous Individuals: Analysis of 1340 Individuals"

_nutrients, 2021, doi:10.3390/nu13092964_

Round 1

Reviewer 1 Report 

In my opinion, the paper is interesting. The number of patients is large. The study is well conducted, providing new data on a much-discussed topic.  The observation that <<....after exclusion of individuals with inflammation (with overweight/obesity and elevated hs-CRP levels), the actual iron deficiency was assessed and was not higher among vegetarians, except in women with regular menstrual cycles.>> is of particular interest for the general discussion concerning this argument. In fact, many data from the literature provide different results. The exclusion of individuals with inflammation adds originality to the study design increasing the power of the results  

Author Response

Thanks for the review.

Reviewer 2 Report

The manuscript “Iron Deficiency in Vegetarian and Omnivorous Individuals: Analysis Of 1340 Individuals” evaluates the prevalence of iron deficiency in vegetarian and omnivorous individuals in both sexes, considering the presence of conditions that elevate serum ferritin (which can mask iron deficiency), such as obesity, insulin resistance (assessed by the homeostatic model of insulin resistance), and inflammation, assessed by measurement of high-sensitivity CRP. The authors show that nutritional status and inflammation affect serum ferritin levels, and this interferes with the correct diagnosis of iron deficiency in both vegetarian and omnivorous individuals. Moreover, they show that women who do not menstruate and men had the same prevalence of iron deficiency when following an omnivorous diet. Among women who menstruate, vegetarians had a higher prevalence of iron deficiency than did omnivores. The manuscript is well-written and needs only minor language revisions from a native English speaker.

Specific comments: 1. In the description of their sample, the authors say that their patients were treated at a private clinic in Sao Paulo. Treated for what? Was this a convenience sample or a true random sample?

  1. In the description of their sample, they mention twice that patients with inflammatory bowel diseases were excluded (lines 84 and 87, page 2). Please, remove the second mention.
  2. Why were patients of Asian ethnicity excluded?
  3. The definition of semi-vegetarians (individuals who eat white meat in up to 3 meals per week) is questionable.
  4. The definitions of low moderate and high hs-CRP is given twice, i.e., in section 2.5 and again at the end of section 3.2. Please, remove the second mention.
  5. Adding up the number of men, women with and without menses in Table 3 gives a total number of 1310 (412+223+675) and not 1340, which is the total sample size. Please, explain the difference.
  6. In line 383, page 11, please change the word hider to the correct hinder.

Author Response

Response to Reviewer

The manuscript was reviewed for American Journal Experts.

Specific comments: 1. In the description of their sample, the authors say that their patients were treated at a private clinic in Sao Paulo. Treated for what? Was this a convenience sample or a true random sample?

The patients came to our office for a general metabolic assessment, without specific health problem, and the sample was randomly taken.

In the description of their sample, they mention twice that patients with inflammatory bowel diseases were excluded (lines 84 and 87, page 2). Please, remove the second mention.

Before:

2.2. Sample

Data from patients aged 18 to 60 years of both sexes treated at a private clinic in the city of São Paulo (Brazil) between 2008 and 2018 were included in this study. Pregnant or nursing women; patients using medications or supplements in the last 6 months able to influence the evaluated variables; patients undergoing nutritional intervention in the last 6 months; smokers; patients with alcohol consumption > 3 times per week; practitioners of physical activity > 3 times a week; patients diagnosed with diseases capable of influencing the evaluated variables (thalassemia, liver disease, hypo- or hyperthyroidism, inflammatory bowel disease, hemochromatosis, cancer with possible metabolic repercussions, rheumatic and autoimmune diseases with systemic inflammatory responses, inflammatory bowel diseases and irritable bowel syndrome) and gastrointestinal tract surgeries that affect nutrient absorption (total or partial gastrectomy, gastric bypass, gastroduodenopancreatectomy and bariatric surgery);

After:

2.2. Sample

Data from patients aged 18 to 60 years of both sexes treated at a private clinic in the city of São Paulo (Brazil) between 2008 and 2018 were included in this study. Pregnant or nursing women; patients using medications or supplements in the last 6 months able to influence the evaluated variables; patients undergoing nutritional intervention in the last 6 months; smokers; patients with alcohol consumption > 3 times per week; practitioners of physical activity > 3 times a week; patients diagnosed with diseases capable of influencing the evaluated variables (thalassemia, liver disease, hypo- or hyperthyroidism, inflammatory bowel disease, hemochromatosis, cancer with possible metabolic repercussions, rheumatic and autoimmune diseases with systemic inflammatory responses, irritable bowel syndrome and gastrointestinal tract surgeries that affect nutrient absorption (total or partial gastrectomy, gastric bypass, gastroduodenopancreatectomy and bariatric surgery);

Why were patients of Asian ethnicity excluded?

Because they have a different proportion between chest and abdomen with upper and lower limbs. So, when they increase body weight, they have more visceral fat, in comparison with general population.

The definition of semi-vegetarians (individuals who eat white meat in up to 3 meals per week) is questionable.

Yes, I agree. But, the literature use this definition in many studies.

The definitions of low moderate and high hs-CRP is given twice, i.e., in section 2.5 and again at the end of section 3.2. Please, remove the second mention.

Before:

Among the different degrees of inflammation, vegetarians had lower ferritin levels than did omnivores, regardless of sex and menstrual blood loss (p <0.0023), except for women who do not menstruate with a low degree of inflammation (p = 0.2633) and for women who menstruate with a moderate degree of inflammation (p = 0.1343). The degrees of inflammation were categorized based on circulating hs-CRP levels into low (≤0.1 mg/dL), moderate (0.1 to 0.3 mg/dL) and high (>0.3 mg/dL).

After:

Among the different degrees of inflammation, vegetarians had lower ferritin levels than did omnivores, regardless of sex and menstrual blood loss (p <0.0023), except for women who do not menstruate with a low degree of inflammation (p = 0.2633) and for women who menstruate with a moderate degree of inflammation (p = 0.1343).

Adding up the number of men, women with and without menses in Table 3 gives a total number of 1310 (412+223+675) and not 1340, which is the total sample size. Please, explain the difference.

There are 29 missing values for iron and 1 for menstrual cycle information.

If you have any questions, please contact me.

In line 383, page 11, please change the word hider to the correct hinder.

Before:

Therefore, it seems important that vegetarian individuals, in comparison to omnivorous individuals, should pay increased attention to their iron nutritional status only when they show blood loss of any origin, by choosing foods with higher concentrations of iron, optimizing factors that improve its absorption and avoiding those that hider it.

After:

Therefore, it seems important that vegetarian individuals, in comparison to omnivorous individuals, should pay increased attention to their iron nutritional status only when they show blood loss of any origin, by choosing foods with higher concentrations of iron, optimizing factors that improve its absorption and avoiding those that hinder it.
